# Non-Invasive Continuous Measurement of Haemodynamic Parameters—Clinical Utility

**DOI:** 10.3390/jcm10214929

**Published:** 2021-10-25

**Authors:** Aleksandra Bodys-Pełka, Maciej Kusztal, Maria Boszko, Renata Główczyńska, Marcin Grabowski

**Affiliations:** 11st Department of Cardiology, Medical University of Warsaw, 02-097 Warsaw, Poland; aleksandra.bodys1@gmail.com (A.B.-P.); maciekrm@gmail.com (M.K.); mariaboszko@gmail.com (M.B.); marcin.grabowski@wum.edu.pl (M.G.); 2Doctoral School, Medical University of Warsaw, 02-091 Warsaw, Poland

**Keywords:** blood pressure wave analysis, continuous non-invasive measurement of haemodynamic parameters, cardiac output

## Abstract

The evaluation and monitoring of patients’ haemodynamic parameters are essential in everyday clinical practice. The application of continuous, non-invasive measurement methods is a relatively recent solution. CNAP, ClearSight and many other technologies have been introduced to the market. The use of these techniques for assessing patient eligibility before cardiac procedures, as well as for intraoperative monitoring is currently being widely investigated. Their numerous advantages, including the simplicity of application, time- and cost-effectiveness, and the limited risk of infection, could enforce their further development and potential utility. However, some limitations and contradictions should also be discussed. The aim of this paper is to briefly describe the new findings, give practical examples of the clinical utility of these methods, compare them with invasive techniques, and review the literature on this subject.

## 1. Introduction

Recently, there has been a rapid development in non-invasive, haemodynamic monitoring technologies. As more and more devices enter the market, the availability of the new approach develops. Despite being often described as the ‘gold standard’, invasive methods can be associated with more complex procedures and the risk of complications, including infection. Therefore, the search for other solutions is necessary. The monitoring of haemodynamic parameters, especially cardiac output, is an essential element of clinical practice. Haemodynamics monitoring can be divided into three groups: invasive methods (requiring the insertion of a specific catheter, as well as direct cardiac and vascular access), less invasive methods (requiring arterial and venous access) and non-invasive methods (without disrupting a patient’s skin and tissues) [1].

This paper aims to analyse the commonly used techniques of haemodynamic parameters’ measurements and to describe the possibilities of currently available non-invasive methods.

## 2. Invasive Methods—The Gold Standard

The accuracy of the non-invasive haemodynamic parameters’ measurements and their comparison to values obtained by the invasive methods, is crucial in clinical practice.

The most invasive, as well as the most precise, technique of haemodynamic parameters’ measurements is pulmonary artery catheterisation (PAC), using the Swan-Ganz catheter. It is inserted via the subclavian or internal jugular vein access and sequentially passes through the venous system into the right atrium of the right ventricle. Next, it is placed in the distal part of the pulmonary artery’s branch. At this location direct measurements can be made, including the cardiac output (CO), central venous pressure (CVP), right atrial and ventricular pressure, pulmonary artery pressure and pulmonary capillary wedge pressure (PCWP). The cardiac output is assessed using the thermodilution technique. A cold saline solution of a known volume and temperature is injected into the right atrium. Passing through the ventricle and into the pulmonary artery, the injectate mixes with the blood, cooling it. Then, a thermistor located at the catheter’s tip measures the blood temperature. Taking into account the temperature and volume of the saline solution, as well as the quantified change in blood temperature, a computer determines the thermodilution profile and calculates the right ventricular cardiac output. The procedure is often repeated and the measurement is averaged. Additionally, PAC enables the indirect measurements of many other haemodynamic parameters such as systemic vascular resistance (SVR), stroke index (SI), pulmonary vascular resistance (Figure 1). Another advantage worth mentioning is the opportunity of obtaining mixed venous oxygen saturation (SvO_2_). This parameter allows for the indirect estimation of hypoxia and peripheral perfusion. Values below 65% are considered to be a sign of increased tissue oxygen consumption. This procedure is associated with the risk of severe complications, such as pulmonary artery dissection, right bundle branch block, and catheter-related infection [2]. Despite its invasive character and more infrequent application, PAC remains the gold standard of CO measurement and is a useful tool in monitoring patients in serious and critical conditions [3]. The use of PAC is recommended in patients with refractory shock and right ventricular dysfunction, as well as patients with severe shock, especially in the case of associated acute respiratory distress syndrome [4].

## 3. Less Invasive Measurement Methods

A slightly less invasive way of obtaining haemodynamic parameters is a method using PiCCO (Pulse Contour Cardiac Output) technology (Gentige, Göteborg, Sweden). It combines a pulse wave contour analysis, the transpulmonary thermodilution method, as well as a venous blood saturation measurement. Thermodilution calibrates the pulse contour analysis in the individual patient. Two vascular accesses are necessary to perform the measurements: central venous access and arterial access via the femoral artery or, optionally, the axillary or brachial artery. A pulse wave contour analysis allows the marking of the CO, mean arterial pressure (MAP), stroke volume (SV), stroke volume variation (SVV), pulse pressure variation (PPV) and SVR values. On the other hand, using transpulmonary thermodilution enables the calculation of the CO, cardiac function (CFI), total end-diastolic volume, volume of excess extravascular water (EVLW) and complete stroke volume. However, oximetry provides several crucial parameters regarding the body’s oxygen management, including venous blood saturation in the superior vena cava, tissue oxygen delivery (DO_2_) and oxygen consumption (VO_2_) [5]. Despite its less invasive character, when compared with PAC, PiCCO is associated with a risk of iatrogenic complications connected with establishing vascular access such as pneumothorax, bleeding, catheter-associated infection or venous thrombosis. Furthermore, the limitations of the pulse wave contour analysis in the case of arrhythmia or the ventricular function-supporting devices should also be noted [6].

ProAQT (Gentige, Göteborg, Sweden) is also used for waveform analysis. However, it does not rely on the thermodilution method and is generally less complex. It can be easily used for both femoral and radial accesses. Nevertheless, when compared with other methods, its inaccuracy and inferiority to thermodilution-based CO and SI measurements is noteworthy [7,8]. Additionally, ProAQT is recognised as inequivalent to the esophageal Doppler system for haemodynamic monitoring during non-vascular, intermediate-risk abdominal surgeries [9]. Recently, Hoppe P et al. used this device to determine the diagnostic accuracy of a mobile application for a snapshot pulse wave analysis in patients undergoing major abdominal surgeries [10].

Another device allowing a less invasive haemodynamic parameters measurement is FloTrac™/Vigileo monitor™ (Edwards, Irvine, CA, USA). It is implemented via arterial access. The FloTrac system enabled the measuring of parameters such as: CO, SV, SVV, MAP and SVR. They were calculated based on 20 s measurements of SV and pulse pressure, which allowed for an almost real-time recording. The results obtained using the FloTrac device in cardiac surgery patients were comparable with those received via an invasive measurement with PAC [11]. One of the limitations of this device is its inability to precisely measure CO in extremely obese patients, as well as during liver transplant surgery and abdominal aortic aneurysm repair surgery [2]. Moreover, the inaccuracy of the FloTrac device measurements is shown in patients with a low CO and high SVR [12].

Lastly, LiDCO (Lithium dilution cardiac output) technology (LIDCO, London, UK), which uses pulse wave analysis, is another less invasive technique. The LiDCO system requires a peripheral artery and central venous catheter (alternatively, a peripheral catheter). The calibration is conducted using the lithium oxide dilution method and is recommended every 8 h. Using this device, the following parameters can be recorded: CO, HR, MAP, CI, SV, SVR, SVRI, SVV, PPV, SPV (systolic blood pressure variations), DO_2_ and DO_2_I (oxygen delivery) [1].

## 4. The Continuous, Non-Invasive Measurement of Blood Pressure and Haemodynamic Parameters

The continuous, non-invasive blood pressure measurement and the monitoring of haemodynamic parameters are assessed for every heart contraction (beat-to-beat mode). These procedures are of an entirely non-invasive character, meaning that they do not disrupt the skin or tissues. Nowadays, the market offers two systems: CNAP (CNSystems Medizintechnik AG, Graz, Austria) and ClearSight (Edwards, Irvine, CA, USA).

When applying these devices, one should start with entering the patient’s data such as sex, age, height and weight. The CNAP measurement requires two cuffs: one on the arm, and the other, a double cuff, on the index and middle finger. A plethysmographic sensor records the blood volume change in the finger. This allows the maintaining of the appropriate pressure in the arm’s cuff in order to sustain a consistent blood volume in the finger. This way, the pressure measured on the arm corresponds to the arterial blood pressure of every heartbeat, and the monitor shows its high-resolution dynamic curve in real time. Naturally, every measurement requires a prior device calibration with a standard oscillometric method (NIPC—Non-Invasive Pulse Co-oximeter) [1,13]. However, the measurement via ClearSight involves applying a specialized cuff on the patient’s finger and is based on a non-invasive pressure measurement on the finger’s artery, using a continuously modified vessel clamp method (so called volume clamp). After eight hours of constant monitoring on one finger, the cuff should be switched to another finger. The total measurement time should not exceed 72 h of constant monitoring. To increase the patient’s comfort, two cuffs can be simultaneously put on in order to alternately measure the parameters between the two fingers. Therefore, an uninterrupted and constant monitoring for up to 72 h is possible [6,11].

Non-invasive measurements allow the assessment of the following (Figure 2):

These values are calculated from the continuously registered pressure curve. Nevertheless, the absolute values are adjusted by the oscillometric measurement.

## 5. Other Non-Invasive Monitoring Methods

NICCOMO (Medis, Ilmenau, Germany) uses thoracic electrical bioimpedance (TEB) to calculate CO. A low amperage and high frequency electricity is transmitted through the chest. During systole, the increase in blood volume in the thorax lowers the resistance of the passing electricity. The impedance is measured via the electrodes placed along the electricity current. Several other parameters can be measured: CI, SV, SI, SVR and SVRI, oxygen delivery index (DO_2_I), oxygen saturation (SpO_2_), HR, SVV, and non-invasive blood pressure (NIBP). Early studies present this technique as an alternative method of haemodynamic monitoring, since it is shown to provide similar values to those obtained via the pulmonary thermodilution method [14].

In patients supported by mechanical ventilation, NICO^TM^ (Non-Invasive Cardiac Output) can be applied [15,16]. The system allows for the non-invasive measurement of CO and ventilation parameters. The device uses a partial, periodic carbon dioxide rebreathing technique, causing a CO_2_ disturbance. Then, CO is calculated using the Fick CO_2_ equation. The values of CO, SV, and pulmonary capillary blood flow (PCBF) can be assessed. NICO measurements correspond relatively well with those obtained from using the thermodilution method [17].

Several advantages of the non-invasive methods include: the patient’s safety, simplicity of application, shorter time until intervention, savings on expensive disposable materials used in invasive methods, and no risk of infection. The relatively limited accessibility of these devices could be considered a disadvantage. Using non-invasive measurements is contraindicated in the following situations: peripheral artery disease, diminished peripheral blood perfusion (e.g., hypothermia), frequent arrhythmias, vascular implants in upper limbs, tremor, and no guarantee of reliable readings in haemodynamically unstable patients [13,18].

A comprehensive comparison of invasive and non-invasive methods is presented in Table 1.

## 6. Assessing Precision and Accuracy

A study involving 21 patients after cardiac surgery, in whom CO was measured up to two hours after the procedure using PiCCO, FloTrac and LiDCO, in comparison with PAC, showed LiDCO as the most precise of the less invasive methods [19].

Regarding the non-invasive methods of parameters measurement, two systems are currently available on the market: CNAP (CNSystems Medizintechnik AG, Graz, Austria) and ClearSight (Edwards, Irvine, CA, USA). Notably, both are based on the volume-clamp method by Penaza [2]. This method was developed in 1969 by Jana Penaza and involves the constant regulation of the cuff’s pressure, based on the plethysmographic visual signal [20].

The ClearSight system was compared with invasive methods in 10 trials with a population of 365 patients. The vast majority of authors showed a good correlation of CO and blood pressure values obtained by a non-invasive measurement with the ‘gold standard’; however, these studies did not meet the FDA’s (Food and Drug Administration) clinical interchangeability criteria [21]. These observations could be explained by the significant diversity of the studied groups and variances in methodology.

It is worth mentioning that the non-invasive measurements were the most accurate in patients with high CO and low SVRI values, and the least accurate in patients with low CO and high SVRI values [22]. The ClearSight, device-mediated cardiac output measurements showed a moderate correlation with the results measured in echocardiography [23].

Conversely, SBP, DBP and MAP measurements made by a CNAP device in a study of 2019, were considered as comparable, obtaining invasive measurements and meeting the criteria for their interchangeable use in stable patients remaining in an intensive care unit after cardiac surgery [24]. Similar results regarding SBP, DBP and MAP measurements were published in a study comparing both methods in patients undergoing elective surgical procedures [25,26,27,28]. The data regarding the accuracy of CO measurements by CNAP is limited. Wagner et al. concluded that the continuous, noninvasive determination of cardiac output is feasible in critically ill individuals [29]. The measurements by CNAP in comparison with those obtained invasively using transpulmonary thermodilution (PiCCO) showed a percentage error of 25%, recognized as an acceptable agreement between the investigated techniques. In another study, CNAP-derived cardiac index measurements were recognized as noninterchangeable with those obtained using 3-dimensional images [30]. However, the high systemic vascular resistance index in patients undergoing abdominal aortic aneurysm surgery may partially account for the observed inaccuracies.

However, a meta-analysis from 2014 of 28 studies, with 919 participants, involving various non-invasive haemodynamic monitoring devices showed inaccuracy and a lack of precision in SBP, DBP and MAP measurements, when compared with invasive monitoring [31]. In response to the mentioned meta-analysis, there were other research discrepancies in implementing the precision and accuracy criteria of the Association for the Advancement of Medical Instrumentation, in relation to blood pressure measurements via non-invasive devices and in a population not fully matching the set assumptions [32]. These comments express the need for establishing new standards regarding the evaluation of non-invasive, haemodynamic parameters measurement methods.

## 7. Non-Invasive Haemodynamic Monitoring—Examples of Clinical Application

When the first devices for non-invasive haemodynamic monitoring entered the market, the research on their clinical application was initiated. Intensive care units and anesthesiologists became the main beneficiaries of the new devices, as they used them for intraoperative cardiovascular monitoring. Recently, TEB provided additional information regarding the haemodynamic alternations resulting from the induction of general anaesthesia [33]. Nonetheless, Hong J Y et al. evaluated the effect of preoperative epidural analgesia on intraoperative cardiovascular parameters during laparoscopic hysterectomy using NICO [34].

Furthermore, non-invasive monitoring enabled the observation of characteristic deviations in haemodynamic parameters in particular groups. A noticeable difference between SBP and DBP, as well as a high acceleration on the pulse wave sigmograph, were the characteristic features of aortic regurgitation (Figure 3A). In cirrhotic patients, we noticed a low peripheral vascular resistance and an increased cardiac output, which were present at rest, as shown in Figure 3B,C. This stems from a systemic vascular vasodilatation and blood redistribution into the visceral vessels. Other departments have also benefited from the technology development.

Although some of the new techniques seem promising for the paediatric population, data regarding this group of patients remains limited [35,36].

## 8. Non-Invasive Haemodynamic Monitoring and Non-Cardiac Surgeries

In surgical patients, haemodynamic instability can occur during the perioperative period, which is caused by the shift in the volume of the intravascular fluid, anaesthetics, and surgical intervention. Therefore, the fundamental aim of anaesthetic monitoring during surgical procedures is to control the haemodynamic parameters. The good credibility and equivalence of the haemodynamic measurements (SBP, DBP, MAP, PPV) obtained in the operating theatre via a non-invasive method, when compared to invasive methods, is noteworthy [37,38,39].

Using a constant, non-invasive arterial blood pressure measurement in patients with ClearSight under general anaesthesia undergoing non-cardiac surgeries decreased the hypotension time by half [40]. These results remain consistent with another study, in which a constant ClearSight monitoring contributed to an earlier diagnosis of hypotension and introduced an effective treatment compared to the standard oscillometric measurement in patients during and after orthopaedic surgery [41]. It is worth stressing, that even a short intraoperative episode of hypotension can greatly affect organ functioning and subsequent complications. In extreme cases, a critical decrease in blood pressure may lead to a sudden cardiac arrest.

In another study, with patients undergoing complete hip or knee joint replacement surgeries, the fluid therapy scheme in the studied group was based on non-invasive haemodynamic monitoring, especially the pulse pressure variation (PPV) and, in the control group, fluid therapy was based on a standard oscillometric blood pressure measurement, every 5 min. The intraoperative fluid therapy under CNAP control resulted in a decreased number of postoperative complications (83% in the control vs. 55% in the studied group) and necessary blood products transfusion (75% patients in the control vs. 38% in the studied group). Moreover, the number of intraoperative hypotension episodes decreased by 33% in the studied group with a lower postoperative mean arterial blood pressure (MAP = 103 mmHg) compared to the control group (MAP = 118 mmHg) [42]. CNAP monitor was also used for the early detection of rapid decreases in blood pressure, occurring during the c-section in patients under subarachnoid anaesthesia [43,44]. Studies stress the usefulness of PPV monitoring in patients under general anaesthesia, in providing haemodynamic stability and showing PPV as an accurate parameter of the haemodynamic response to fluid therapy [45,46].

Furthermore, a high correlation of non-invasive PPV measurements, when compared with invasive methods, is shown [47,48]. These results suggest that non-invasive haemodynamic monitoring is a valuable element of intraoperative monitoring; however, its correlation with patients’ prognoses requires further research [41]. There are limited available data regarding the clinical implications of using the NICO system in non-cardiac patients [49].

Moreover, there is no direct comparison of clinical applications in both non-invasive monitoring systems, CNAP and ClearSight.

## 9. Non-Invasive Haemodynamic Monitoring and Cardiosurgery and Interventional Cardiology

Another important group of patients, in whom haemodynamic monitoring is essential, are patients undergoing cardiosurgical procedures. In haemodynamically stable patients undergoing coronary artery bypass grafting (CABG), postoperative cardiac output values measured by a non-invasive technique were comparable to those measured via an invasive PiCCO method [50,51] and those calculated based on a transthoracic echocardiography [52]. Lorsomradee S. et al., in a group of 36 patients, compared the CO values acquired via a non-invasive and an invasive method and obtained similar results [50]. Similar conclusions were drawn in Bronch O. et al.’s study, where, in 40 patients undergoing CABG, the CO values were measured while inducing a general anaesthesia until discharge from the intensive care unit [51]. However, these results were not reflected in other studies, in which non-invasive cardiac output measurements after cardiac surgery, despite a good trend of real-time CO changes, did not meet the equivalence criteria of both methods, when compared with the invasive methods [53,54]. Noteworthy, blood pressure measurements in these studies were credible, comparable and met the equivalence criteria with regard to both invasive methods.

In a study of 33 patients with severe aortic stenosis undergoing percutaneous aortic valve implantation (TAVI) via transfemoral access, no significant differences were found in the accuracy of blood pressure measurements using a non-invasive method (CNAP) and an invasive (intra-arterial) method. Furthermore, the non-invasive blood pressure measurement during fast heart stimulation, accurately and immediately showed significant alternations in the haemodynamic parameters [55]. Similar results in accuracy of blood pressure and cardiac output were obtained in two studies in which the ClearSight system was applied and compared to the invasive monitoring in patients with aortic and mitral valve replacement [56,57]. Another study found that an agreement between NICO and invasive hemodynamic monitoring was clinically acceptable, but had a tendency to underestimate CO compared to the termodilution method [49].

Apart from these findings, there are very limited data showing any clinical benefits, including patient outcomes and cost effectiveness, from using non-invasive hemodynamic monitoring.

## 10. Non-Invasive Haemodynamic Monitoring and Intensive Care

Haemodynamic monitoring is essential for patients in critical condition and for those requiring intensive care. The studies regarding the utility of non-invasive haemodynamic monitoring methods in this population gave contradictory conclusions. The research involving 40 intensive care unit patients showed an accuracy of non-invasive monitoring blood pressure measurements when compared to the invasive measurements [58]. Another study on 55 intensive therapy patients proved the precision of non-invasive DBP and MAP measurements via CNAP. These values were similar to those acquired via an invasive method. However, the SBP measurements were less precise and accurate when compared with the values from the intra-arterial catheter [59]. By contrast, smaller studies involving intensive therapy patients questioned the accuracy of non-invasive measurements with both CNAP and ClearSight and their application in the studied population [22,60,61]. It is worth mentioning that the studied groups consisted of patients with a broad spectrum of underlying diseases, contributing to their serious conditions. Again, there was a small number of studies regarding the NICO system application in intensive care units with one study showing a moderate correlation with invasive methods, but which was still applicable to patients not breathing spontaneously [62].

Regarding cardiology, non-invasive haemodynamic monitoring techniques were tested on a group of 84 heart failure patients, mainly NYHA III and IV; the mean ejection fraction was 27%. The non-invasive measurement of cardiac output compared to the thermodilution method was revealed to be overestimating and not recommended for this population [63]. Despite being disappointing, this result remains in line with observations from other studies, in which non-invasive haemodynamic monitoring methods showed less precise measurements in the case of decreased cardiac output and increased peripheral vascular resistance [22]. It might also partially explain the inaccuracy of measurements for patients in critical conditions that remain in intensive care units.

The clinical utility of NICCOMO in intensive care unit patients requires further investigation as the data regarding this matter are limited [64].

## 11. Summary

Considering the complexity of procedures and the possible severe complications associated with invasive haemodynamic monitoring, alternative solutions are being widely explored. Non-invasive methods were found to be successful in numerous areas. In surgical patients, they were used for intraoperative cardiovascular monitoring and the early detection of haemodynamic alternations, as well as for adjusting the fluid therapy. Furthermore, a reduction in postsurgical complications was noted. These findings come from various hospital wards, other than the cardiac department, including anaesthesiology, orthopaedic surgery and gynaecology wards. Even though a wide range of parameters can be obtained, in some cases an invasive approach is necessary. This includes not only critically ill patients, but also those in shock or suffering from heart failure. Nevertheless, in some patients, the implementation of the non-invasive methods might be the missing link between an invasive approach, such as thermodilution, and a simple oscillometric cuff measurement. Hence, patients with non-cardiac conditions which severely alter their hemodynamic state, e.g., cirrhosis and chronic kidney disease, should be the next potential group that could benefit from non-invasive hemodynamic monitoring and require further investigation.

The data regarding monitoring patients in intensive care units, using new technologies, remain contradictory and require more studies. Moreover, it is crucial to further investigate how these non-invasive techniques could influence the therapeutic decision-making process and patients’ prognoses, as well as analyse the economic aspects of these interventions. In addition, other potential clinical practice applications should also be widely explored. Furthermore, a substantial need for establishing and standardizing the evaluation criteria of non-invasive haemodynamic monitoring devices is noted. The current criteria were initially applied only to oscillometric temporary measurements. Therefore, they seem to not be applicable in evaluating the constant, non-invasive measurements of the haemodynamic parameters [2,32].

Finally, the approach to choose the right method should always be individually tailored to the patient and still recognize the contradictions, as well as the limitations.

## Figures and Tables

**Figure 1 jcm-10-04929-f001:**
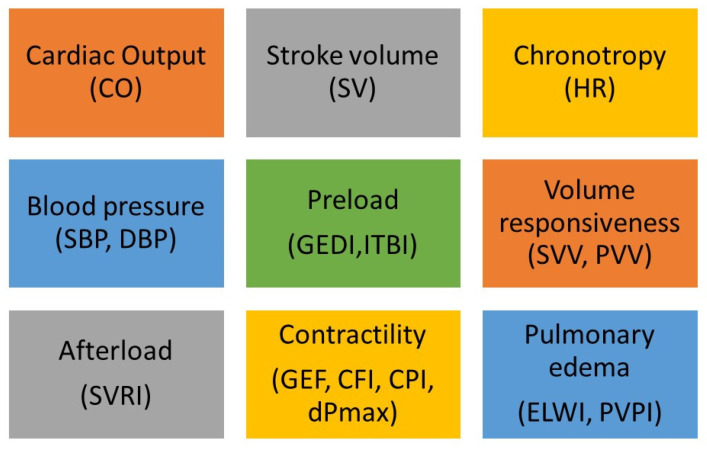
Haemodynamic parameters measured with invasive methods. Global end-diastolic index (GEDI); extravascular lung water index (ELWI); cardiac lung water index (ELWI), cardiac function index (CFI), global ejection fraction (GEF); continuous left ventricular contractility (dPmx), Pulmonary vascular permeability index (PVPI), systemic vascular resistance index (SVRI).

**Figure 2 jcm-10-04929-f002:**
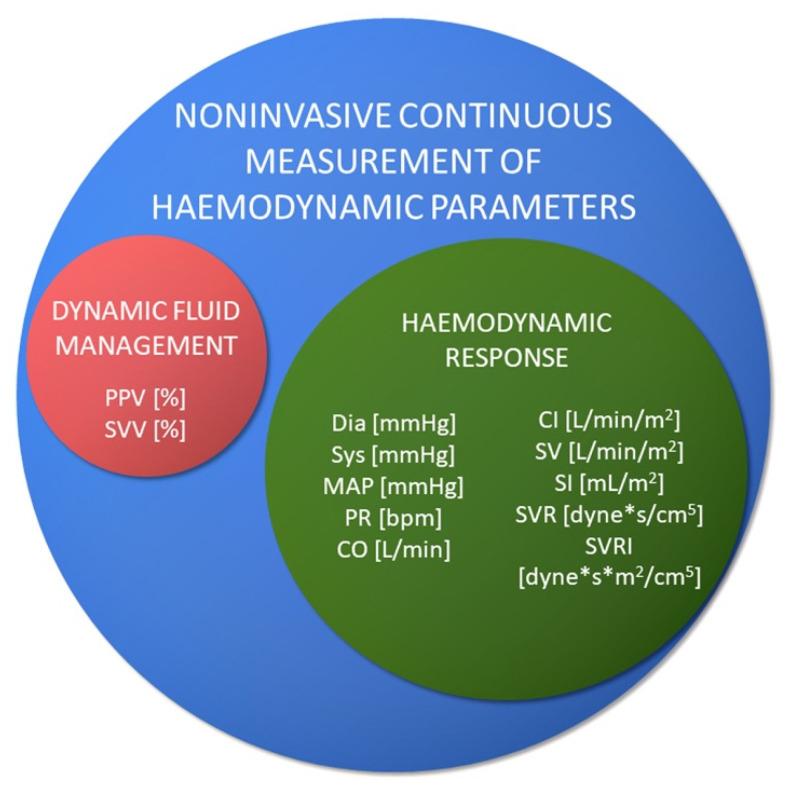
Haemodynamic parameters measured with non-invasive methods and their units. Dia—diastolic arterial pressure (DBP); Sys—systolic arterial pressure (SBP); MAP—mean arterial pressure; PR—pulse rate, heart rate (BMP); CO—cardiac output; CI—cardiac index; SV—stroke volume; SI—stroke index; SVR—systemic vascular resistance; SVRI—systemic vascular resistance index; PPV—pulse pressure variation; SVV—stroke volume variation.

**Figure 3 jcm-10-04929-f003:**
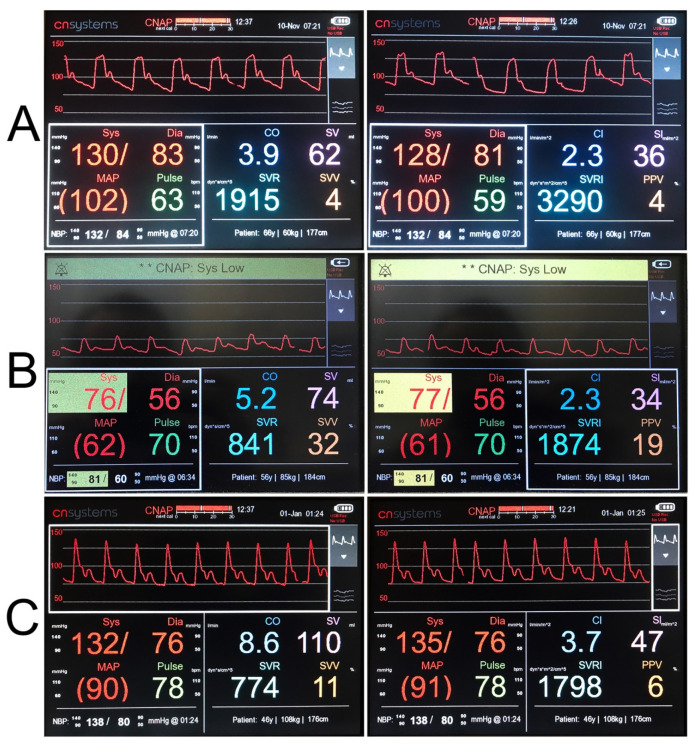
Record of measured haemodynamic parameters in patients (example). (**A**) A 66-year-old patient with severe aortic regurgitation was admitted to cardiology department to assess the width and qualification of the ascending aorta for surgical valve treatment. Echocardiography demonstrated normal heart size, distended ascending aorta, and aortic bulb. Severe aortic regurgitation. EF 59%. Analysis of the pulse wave shown in the diagram reveals a well-defined dicrotic notch that is characteristic of aortic regurgitation. (**B**) A 55-year-old patient with restrictive cardiomyopathy caused by genetically determined transthyretin amyloidosis presenting to the hospital for re-evaluation of indications of combined liver and heart transplant. This disease is characterized by left ventricular diastolic dysfunction. The figure shows characteristic low cardiac output and stroke volume with high systemic vascular resistance. (**C**) A 46-year-old male with mixed HBV/ALD aetiology cirrhosis, complicated with hepatic encephalopathy and ascites, with a history of portal hypertension, oesophageal varices bleeding, arterial hypertension, heart failure, asthma, and type 2 diabetes. Qualified for liver transplant surgery. This example perfectly shows the features of hyperdynamic circulation characteristic of hepatic cardiomyopathy: low systemic vascular resistance, high cardiac output, high stroke volume and tachycardia. Sys—systolic blood pressure; Dia—diastolic blood pressure; MAP—mean arterial pressure, CO—cardiac output; SV—stroke volume; SVR—systemic vascular resistance; PPV—pulse pressure variation; CI—cardiac index; SI—stroke volume index; SVRI—systemic vascular resistance index; SVV—stroke volume variation.

**Table 1 jcm-10-04929-t001:** Comparison of invasive and non-invasive methods.

	Invasive Methods	Non-Invasive Methods
Complexity of application	Necessity of complex procedures	Simple application
Risk of complications	Recognizable/considerable	Limited
Possibility of continuous measurement	Yes	Yes (in some cases)
Need for calibration	Yes, depends on device type	Yes
Accuracy and precision	The gold standard	Comparable (in some cases, depending on measured parameter and applied criteria)
Time until intervention	Longer	Shorter
Fit for monitoring critically ill patients	Yes	No
Fit for intra-operative monitoring	Yes	Yes
Availability	Common	Limited, but growing
Indications	Patients with refractory shock and right ventricular dysfunction Patients with severe shock and acute respiratory distress syndrome In some cases, to differentiate cardiogenic pulmonary edema from non-cardiogenic	Patients who are out of the critical stage Patients undergoing elective procedures Patients who are at risk of haemodynamic compromise or where the invasive methods put patients at unnecessary, increased risk
Contradictions	Tricuspid or pulmonary walve prosthetisis which can be damage Infective endocarditis of the tricuspid or pulmonary valve Severe tricuspid or pulmonic stenosis Right heart mass (tumor or clot) Patients with coagulopathy	Peripheral artery disease Diminished peripheral blood perfusion (ex. hypothermia), Frequent arrhythmias Vascular implants in upper limbs, Tremor
Limitations	life-threatening arrhythmias	Patients critically ill/in shock
Complications	Haemothorax, pneumothorax Atrial fibrillation Ventricular arrhythmia Thromboembolic events Damage to the valves	These procedures are non-invasive

## Data Availability

Not applicable.

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
