# Peer review of "Non-Invasive Continuous Measurement of Haemodynamic Parameters—Clinical Utility"

_jcm, 2021, doi:10.3390/jcm10214929_

Round 1
Reviewer 1 Report
This review article describes an interesting topic, which evaluates the effectiveness and usefulness of non-invasive hemodynamic measurement. The article concludes the most important knowledge regarding this topic.
I have some minor comments which needs to be answered and corrected before publication:
line 38 - in one sentence you use the term "pulmonary artery catheter" and in the next sentence you use the term "Swan-Ganz catheter". Please choose one of them and use it uniformly.
line 39: please replace "internal carotid vein" to internal jugular vein
Please clear out which are the exact indications of pulmonary artery catheter (for example hemodynamic instability with right heart failure, heart transplantation etc.). Check ESC guidelines and ESICM guidelines (Cecconi et al, Intensive Care Med, 2014;40:1795-815.
You wrote that using non-invasive measurement is contraindicated when body-mass is between 40-180 kg. Please, correct it.
Table 1: it is not true that invasive methods do not need calibration - for example measurement with PiCCO needs calibration. Please be more precise.
Indicate the disadvantages of non-invasive methods in more details.
Give more information about the indications and usage of invasive and non-invasive methods.
Author Response
Dear Reviewers,
Thank you for the discerning comments concerning our manuscript entitled “Non-invasive continuous measurement of hemodynamic parameters - clinical utility” which we accept with comprehension and gratitude. We have studied your comments carefully and made corrections which we hope will meet with your approval. Your questions or comments are answered in detail below, with original reviewer comments denoted in boldface, our responses in regular typeface and all changes in the manuscript in red font.
We would like to thank you for kind words and we corrected the manuscript according to your advices.
Responses to reviewers:
Reviewer 1
line 38 - in one sentence you use the term "pulmonary artery catheter" and in the next sentence you use the term "Swan-Ganz catheter". Please choose one of them and use it uniformly.
Response: We agree.
Change: We use only one term in the paper
line 39: please replace "internal carotid vein" to internal jugular vein
Response: Thank you for the advice
Change: As you suggested we use internal jugular vein
Please clear out which are the exact indications of pulmonary artery catheter (for example hemodynamic instability with right heart failure, heart transplantation etc.). Check ESC guidelines and ESICM guidelines (Cecconi et al, Intensive Care Med, 2014;40:1795-815.
Response: We agree
Change: We added more indications in the manuscript.
You wrote that using non-invasive measurement is contraindicated when body-mass is between 40-180 kg. Please, correct it.
Response: We agree
Change: This part is deleted from contraindication
Table 1: it is not true that invasive methods do not need calibration - for example measurement with PiCCO needs calibration. Please be more precise.
Response: We agree
Change: We changed accordingly the table
Indicate the disadvantages of non-invasive methods in more details.
Response: We agree
Change: More disadvantages have been added to the table
Give more information about the indications and usage of invasive and non-invasive methods.
Response: Thank you for your advice
Change: More information about it have been added.
Reviewer 2 Report
The authors wrote a review on Non-invasive hemodynamic monitoring, which is a subject worth a good review. It is narrative review, without any tables summarising physiological and/or clinical studies.
First, different techniques/platforms of invasive/less-invasive/non-invasive techniques are described with varying levels of detail, before giving examples and discussing clinical application of predominantly one technique (CNAP).
General comment: It would be good to have the language corrected by a native English speaker.
Comments chapter by chapter:
Abstract
I am not so sure wether these monitoring devices are useful for evaluating patients pre-operatively.
Introduction
- Please rephrase the first sentence.
- “Cardiovascular assessment”, ”techniques assessing the cardiovascular system”, could better be changed to "hemodynamic monitoring"
Invasive methods
- Carotid vein? => Internal jugular vein
- I would add some words on how cardiac output measurements work using Swan ganz catheters (termodilution technique), and I would add the advantage of obtaining mixed venous oxygen saturation.
Less invasive methods
- I think PiCCO is a registered name (by Getinge), and not the name of a technique, please rephrase the first sentence. Furthermore, I would add in the text that thermodilution calibrates the pulse contour analysis in the individual patient.
- For proAQT, Flotrac etc, I would add the manufacturer as well since these are “trade names” invented by the manufacturers.
Non-invasive methods
- What does NIPC mean?
- The legend for fig 2 seems a bit double..
Other non-invasive:
- What is NICCOMO? Please explain abbreviation or mention company or so ever.
-NICO measurements fairly well correspond
- “Contradiction”should be changed to “contraindication”
Table 1:
- The statement that non-invasive techniques are cost-efficient and invasive are not needs some references as evidence, and I doubt that there is sufficient evidence.
- Assessing precision and accuracy
- Are there any data regarding accuracy of CO measurement by CNAP (you only mention continuous blood pressure measurements)
- Non-invasive haemodynamic monitoring – examples of clinical application
Figure 3, 4, 5
I don’t understand the main message of these patient examples. How does the aortic stenosis (A) or aortic regurgitation (B) manifest in the monitor parameters?
The same applies (to a lesser extent) to fig 4 and fig 5: You should make it clear to the reader what we are looking at, help with interpretation of the measured parameters and why that is so useful to measure in these patients.
Furthermore, all are CNAP recordings. If this aims to be a review on non-invasive hemodynamic monitoring, examples of other techniques could have a place. Alternatively: delete 2 out of the 3 figures.
- CNAP and non-cardiac surgeries’
- Instead of focussing on only CNAP: Are there any clinical data regarding clearsight, NICCOMO or NICO worth mentioning? And if there are none, that should be mentioned, since you aim for a review on non-invasive monitoring in general.
- CNAP and cardiosurgery and interventional cardiology
- See my comment on section 8, what about the other techniques? Furthermore, please note that studies showing a clinical benefit are lacking.
- CNAP and intensive care
- Same here, anything on other non-invasive techniques?
- Non-invasive haemodynamic monitoring– potential future applications
- This is rather short. Do you have any suggestions for future trials, or any emerging techniques? I note that this is better described in the summart section.
Author Response
Dear Reviewers,
Thank you for the discerning comments concerning our manuscript entitled “Non-invasive continuous measurement of hemodynamic parameters - clinical utility” which we accept with comprehension and gratitude. We have studied your comments carefully and made corrections which we hope will meet with your approval. Your questions or comments are answered in detail below, with original reviewer comments denoted in boldface, our responses in regular typeface and all changes in the manuscript in red font.
We would like to thank you for kind words and we corrected the manuscript according to your advices
Reviewer 2
General comment: It would be good to have the language corrected by a native English speaker.
Response: Thank you for your advice
Change: We asked for opinion of native English speaker and applied some alterations in text
Abstract: I am not so sure wether these monitoring devices are useful for evaluating patients pre-operatively.
Response: We agree
Change: We rewrote this sentence:
The use of these techniques for assessing patients’ eligibility before cardiac procedures, as well as for intraoperative monitoring is currently being widely investigated.
We discuss the usefulness of devices later on (for example reference [24]).
Introduction - Please rephrase the first sentence.
Response: Thank you for your advice
Change: We rephrased the sentence as follows
Recently, there has been a rapid development of the non-invasive haemodynamic
monitoring technologies. technologies for non-invasive cardiovascular assessment can
be observed.
Introduction - “Cardiovascular assessment”, ”techniques assessing the cardiovascular system”, could better be changed to "hemodynamic monitoring"
Response: We agree
Change: We changed accordingly to your suggestion
Invasive methods - Carotid vein? => Internal jugular vein
Response: We agree
Change: We changed accordingly to your suggestion
Invasive - I would add some words on how cardiac output measurements work using Swan ganz catheters (termodilution technique), and I would add the advantage of obtaining mixed venous oxygen saturation.
Response: We agree
Change: Added more information as you suggested. Moreover in this part we decided to change the fig.1 to represent not only the CNAP but to represent parameters measured by invasive methods.
Cardiac output is assessed using the thermodilution
technique. A cold saline solution of known volume and temperature is injected into the
right atrium. Passing through the ventricle and into the pulmonary artery the injectate mixes with the blood, cooling it. Then, a thermistor located at the catheter’s tip measures
the blood temperature. Taking into account the temperature and volume of the saline
solution, as well as the quantified change in blood temperature the computer determines
the thermodilution profile and calculates right ventricular cardiac output. The procedure
is often repeated, and the measurement averaged. Another advantage
worth mentioning is the opportunity of obtaining mixed venous oxygen saturation
(SvO2). This parameter allows for indirect estimation of hypoxia and peripheral
perfusion. Values below 65% are considered as a sign of increased tissue oxygen
consumption. It is recommended to use PAC in patients with refractory shock
and right ventricular dysfunction, patients with severe shock especially in the case of
associated acute respiratory distress syndrome.
Less invasive methods - I think PiCCO is a registered name (by Getinge), and not the name of a technique, please rephrase the first sentence. Furthermore, I would add in the text that thermodilution calibrates the pulse contour analysis in the individual patient.
Response: We agree
Change: We explained the acronym and added the name of manufacturer. Moreover, we added a part about thermodilution and its calibration of pulse contour analysis
Less invasive methods - For proAQT, Flotrac etc, I would add the manufacturer as well since these are “trade names” invented by the manufacturers.
Response: We agree.
Change: We added to the trade names their manufacters.
Non-invasive methods - What does NIPC mean?
Response: We agree
Change: We added the full explanation of acronym NIPC (Non-invasive Pulse Co-oximeter)
Non-invasive- The legend for fig 2 seems a bit double
Response: Thank you for your advice
Change: We changed the legend for fig 2
Other non-invasive - What is NICCOMO? Please explain abbreviation or mention company or so ever.
Response: We agree
Change: As it is a device brand name we added name of the manufacturer
Other non-invasive -NICO measurements fairly well correspond
Response: We agree
Change: We modified the statement accordingly to your suggestion
Other non-invasive - “Contradiction” should be changed to “contraindication”
Response: Thank you for your advice
Change: We changed the misspelling
Table 1 - The statement that non-invasive techniques are cost-efficient and invasive are not needs some references as evidence, and I doubt that there is sufficient evidence.
Response: We agree. We checked again for evidence which in fact is limited and not conclusive
Change: Part about cost-efficiency was deleted from the table
Assessing precision and accuracy - Are there any data regarding accuracy of CO measurement by CNAP (you only mention continuous blood pressure measurements)
Response: We agree
Change: We added more information about precision and accuracy of CNAP:
Data regarding accuracy of CO measurement by CNAP is
limited. Wagner et al concluded that continuous noninvasive cardiac output
determination is feasible in critically ill individuals [x]. Measurements by CNAP in
comparison with those obtained invasively using transpulmonary thermodilution (PiCCO)
showed a percentage error of 25 %, recognized as an acceptable agreement between
the investigated techniques. In another study, CNAP- derived cardiac index
measurements were recognized as not interchangeable with those obtained using 3-
dimensional images. However, high systemic vascular resistance index in patients
undergoing abdominal aortic aneurysm surgery may partially account for observed
inaccuracies.
Non-invasive haemodynamic monitoring – examples of clinical application Figure 3, 4, 5
I don’t understand the main message of these patient examples. How does the aortic stenosis (A) or aortic regurgitation (B) manifest in the monitor parameters?
The same applies (to a lesser extent) to fig 4 and fig 5: You should make it clear to the reader what we are looking at, help with interpretation of the measured parameters and why that is so useful to measure in these patients.
Furthermore, all are CNAP recordings. If this aims to be a review on non-invasive hemodynamic monitoring, examples of other techniques could have a place. Alternatively: delete 2 out of the 3 figures.
Response: We agree
Change: We deleted fig 4 and fig 5. Also, we modified fig 3 and added more information about the interpretation of the measured parameters
CNAP and non-cardiac surgeries - Instead of focussing on only CNAP: Are there any clinical data regarding clearsight, NICCOMO or NICO worth mentioning? And if there are none, that should be mentioned, since you aim for a review on non-invasive monitoring in general.
Response: We agree
Change: Part previously describing generally non-invasive monitoring is now more focused on ClearSight device. Added more clinical data about NICO device.
CNAP and cardiosurgery and interventional cardiology - See my comment on section 8, what about the other techniques? Furthermore, please note that studies showing a clinical benefit are lacking.
Response: We agree
Change: Added more information about Clearsight and NICO from new references
Similar results in accuracy of blood pressure and cardiac output were obtained in two studies in which ClearSight system was applied and compared to invasive monitoring in patients with aortic and mitral valve replacement. [54,55]. Another studies found agreement between NICO and invasive hemodynamic monitoring is clinically acceptable, however with a tendency to underestimate CO compared to termodilution method [56].
Apart from these findings there is very limited data showing any clinical benefit
including patients outcomes and cost effectiveness from using non-invasive
hemodynamic monitoring.
CNAP and intensive care - Same here, anything on other non-invasive techniques?
Response: We agree
Change: Added minor alterations to the text. Unfortunately there is not much evidence in this specific group of patients and most of them are conflicting.
Non-invasive haemodynamic monitoring– potential future applications - This is rather short. Do you have any suggestions for future trials, or any emerging techniques? I note that this is better described in the summart section
Response: We agree
Change: We deleted this part and added it to the summary with some new ideas about potential future applications. Another change is funding – our grant from ministry is unfortunately over and this paper will be funded by Medical University of Warsaw.